# Strategies for Efficient Expression of Heterologous Monosaccharide Transporters in *Saccharomyces cerevisiae*

**DOI:** 10.3390/jof8010084

**Published:** 2022-01-15

**Authors:** Marilia M. Knychala, Angela A. dos Santos, Leonardo G. Kretzer, Fernanda Gelsleichter, Maria José Leandro, César Fonseca, Boris U. Stambuk

**Affiliations:** 1Center of Biological Sciences, Department of Biochemistry, Federal University of Santa Catarina, Florianópolis 88040-900, SC, Brazil; marilia.mknychala@gmail.com (M.M.K.); angela.asds@gmail.com (A.A.d.S.); leo_kretzer@hotmail.com (L.G.K.); gelsnanda@gmail.com (F.G.); 2Laboratório Nacional de Energia e Geologia, I.P., Unidade de Bioenergia, Estrada do Paço do Lumiar 22, 1649-038 Lisboa, Portugal; mjose.leandro@itqb.unl.pt (M.J.L.); cesfonseca76@gmail.com (C.F.); 3ITQB NOVA, Instituto de Tecnologia Química e Biológica António Xavier, Av. da República, 2780-157 Oeiras, Portugal; 4Discovery, R&D, Chr. Hansen A/S, 2970 Hørsholm, Denmark

**Keywords:** xylose, *hxt*-null, ubiquitinylation, lysine, truncated permease, endocytosis, membrane protein turnover, *XUT1*, *ROD1*, *ROG3*

## Abstract

In previous work, we developed a *Saccharomyces cerevisiae* strain (DLG-K1) lacking the main monosaccharide transporters (*hxt*-null) and displaying high xylose reductase, xylitol dehydrogenase and xylulokinase activities. This strain proved to be a useful chassis strain to study new glucose/xylose transporters, as SsXUT1 from *Scheffersomyces stipitis*. Proteins with high amino acid sequence similarity (78–80%) to SsXUT1 were identified from *Spathaspora passalidarum* and *Spathaspora arborariae* genomes. The characterization of these putative transporter genes (*SpXUT1* and *SaXUT1*, respectively) was performed in the same chassis strain. Surprisingly, the cloned genes could not restore the ability to grow in several monosaccharides tested (including glucose and xylose), but after being grown in maltose, the uptake of ^14^C-glucose and ^14^C-xylose was detected. While SsXUT1 lacks lysine residues with high ubiquitinylation potential in its N-terminal domain and displays only one in its C-terminal domain, both SpXUT1 and SaXUT1 transporters have several such residues in their C-terminal domains. A truncated version of *SpXUT1* gene, deprived of the respective 3′-end, was cloned in DLG-K1 and allowed growth and fermentation in glucose or xylose. In another approach, two arrestins known to be involved in the ubiquitinylation and endocytosis of sugar transporters (*ROD1* and *ROG3*) were knocked out, but only the *rog3* mutant allowed a significant improvement of growth and fermentation in glucose when either of the XUT permeases were expressed. Therefore, for the efficient heterologous expression of monosaccharide (e.g., glucose/xylose) transporters in *S. cerevisiae*, we propose either the removal of lysines involved in ubiquitinylation and endocytosis or the use of chassis strains hampered in the specific mechanism of membrane protein turnover.

## 1. Introduction

The use of sustainable alternatives to fossil fuels has been growing for the last few decades, ethanol being the primary liquid biofuel in the USA and Brazil with 80% of the global production [1,2,3,4]. While first generation (1G) biofuels rely on food-based plant materials (e.g., sugarcane and corn), second generation (2G) biofuels use lignocellulosic biomasses [5]. Lignocellulosic plant materials, obtained from agricultural and forestry residues, exist in large quantities and are prevalent on earth [4,6]. The most recent technologies developed for the biochemical conversion of lignocellulosic biomass typically involve several steps, including pre-treatment, enzymatic hydrolysis and fermentation, efficient conversion of all the sugars available being a major requirement for an economically feasible 2G bioethanol process [7,8].

Lignocellulosic biomass is composed of cellulose (a linear polymer of glucose molecules linked by β-1,4 glycosidic bounds), hemicellulose (a branched and highly heterogeneous polymer containing both hexoses and pentoses) and lignin. Thus, the resulting biomass hydrolysates have various hexoses and pentoses available, and after glucose, xylose and arabinose are the second and third most abundant sugars in most plant biomass hydrolysates [9,10]. While glucose can be easily fermented by industrial *Saccharomyces cerevisiae* strains [11], this yeast is not naturally able to ferment the pentoses, xylose and arabinose, unless it is genetically modified to express the assimilation routes for these sugars [12,13,14]. In the case of xylose (Figure 1), genes encoding for xylose reductase (XR) and xylitol dehydrogenase (XDH) from *Scheffersomyces stipitis* or *Spathaspora passalidarum*, or xylose isomerase (XI) from bacteria and fungi, have been extensively used [14,15,16,17,18]. Since both pathways transform xylose into xylulose (Figure 1), it is also necessary to overexpress the endogenous xylulokinase (XK) gene that will enhance the entrance of this sugar into the pentose phosphate pathway. While the XI pathway can provide higher ethanol yields, the XR/XDH route allows higher ethanol productivities and higher xylose consumption rates by engineered *S. cerevisiae* yeasts [19,20].

When the metabolic pathway for xylose is established intracellularly, the transport of the sugar across the plasma membrane can become the limiting step in the assimilation and fermentation of this pentose. The uptake of xylose in *S. cerevisiae* is mediated by a large family of hexose transporters, encoded by the *HXT1*-*HXT17* and *GAL2* genes, but is rather inefficient, as the affinities for this pentose are significantly lower than for glucose [12,21]. Thus, contributions to improve 2G ethanol production have been given via the heterologous expression of pentose transporters in *S. cerevisiae* [22,23,24,25,26,27]. Accordingly, the engineering of pentose transporters in recombinant *S. cerevisiae* yeasts is the subject of several recent reviews [28,29,30,31]. A suitable platform to clone and characterize monosaccharide transporters was created in 1999 by deleting all known hexose transporters in *S. cerevisiae*, as this *hxt*^0^-null EBY.VW4000 strain is not able to grow on media with glucose, fructose or mannose as the sole carbon source (although it still grows very slowly on galactose), but grows normally on maltose, as this sugar is transported by different (maltose-specific) permeases encoded by *MALx1* genes [32,33]. This *hxt*^0^-null strain has been used to characterize hundreds of hexose transporters from other yeasts, fungi, plants and other eukaryotes, including human GLUT transporters as well as xylose transporters, after the introduction of a xylose utilization pathway in this *hxt*^0^-null strain [24,34]. One drawback of this *hxt*^0^-null EBY.VW4000 strain is that it required 16 successive deletion rounds with the LoxP/Cre system, which resulted in gene losses and several chromosomal rearrangements, affecting, for example, sporulation and spore germination by this *hxt*^0^-null strain [35]. Nevertheless, a recent publication shows that by using modern genome editing (CRISPR, *C*lustered *R*egularly *I*nterspaced *S*hort *P*alindromic *R*epeats) technology, it is possible to construct an *hxt*^0^-null strain without these drawbacks [36].

Our group has been using a platform *S. cerevisiae* strain (DLG-K1) to study xylose and other sugar transporters. This strain has both: i) the main hexose transporters knocked out (*hxt1-7*Δ and *gal2*Δ) and is thus unable to grow in glucose, fructose and other monosaccharides; and ii) the overexpression of the xylose reductase and xylitol dehydrogenase genes *XYL1* and *XYL2* from *Sc. stipitis*, and the endogenous xylulokinase gene *XKS1* from *S. cerevisiae* (see Figure 1), showing high activities of the respective enzymes. This DLG-K1 strain was used to characterize the xylose fermentation capacity of yeast strains expressing individual *S. cerevisiae* endogenous hexose transporters (e.g., *HXT1*, *HXT2*, *HXT5* and *HXT7*) [21] and allowed the characterization of three novel xylose transporters (*XUT1*, *QUP2* and *HXT2.6*) from *Sc. stipitis* [27]. While the expression of SsXUT1 in DLG-K1 allowed ethanol production from xylose or glucose/xylose mixtures, SsHXT2.6 led to the production of both ethanol and xylitol, and SsQUP2 generated mainly xylitol during xylose consumption. Since the genome of other new xylose-fermenting yeast species (e.g., *Sp. passalidarum* [37,38] or *Sp*. *arborariae* [39,40]) have genes encoding putative transporters with high amino acid sequence similarity (78–80%) to SsXUT1, we decided to clone and characterize these putative permeases in the DLG-K1 platform strain. In the present work, we show that the cloned genes (*SsXUT1* and *SpXUT1*) could not restore the ability of the yeast strain to grow in several monosaccharides tested, although after growth on maltose, ^14^C-glucose and ^14^C-xylose uptake was observed. Thus, we analyzed these transporters in relation to the presence/absence of lysine residues with ubiquitinylation potential in the N- or C-terminal domains, as well as the impact of two α-arrestins known to be involved in sugar transporter ubiquitinylation and endocytosis in the membrane protein turnover of *S. cerevisiae*.

## 2. Materials and Methods

### 2.1. Strains, Media and Growth Conditions

The yeast strains and plasmids used in this study are listed in Table 1. The *S. cerevisiae* DLG-K1 strain was constructed by transforming the *hxt*-null strain KY73 [41] with an integrative plasmid at the *AUR1* loci (plasmid pAUR-XKXDHXR [42]) that has the three genes required for efficient xylose utilization (*XYL1* and *XYL2* from *Sc. stipitis*, and *XKS1* from *S. cerevisiae*) under control of the PGK promoter and terminator, allowing high xylose reductase, xylitol dehydrogenase and xylulokinase activities [21,27]. The *hxt*-null strain KY73 had its main hexose transporter genes (*HXT1* to *HXT7*) knocked out using genes of prototrophic markers, while the gene *GAL2* (encoding a galactose permease capable of transporting several monosaccharides) was deleted using a *URA3* gene flanked by short direct repeats and further selection with 5-fluoro-orotic acid to promote the excision of the *URA3* gene from the *gal2*Δ loci [41]. This *hxt*-null KY73 strain, unable to consume glucose and several other monosaccharides have been used, for example, to characterize amino acid residues important for the specificity and affinity of several yeast hexose transporters [43,44,45], the direct uptake of sucrose by *S. cerevisiae* [46] or even the extracellular hydrolysis of maltotriose by some strains of this yeast [47]. The *Escherichia coli* strain DH5α was used for cloning and was grown in Luria broth (1% tryptone, 0.5% yeast extract, 0.5% sodium chloride, Sigma-Aldrich Brasil Ltd.a., São Paulo, SP, Brazil) supplemented with 100 mg/L ampicillin. Yeasts were grown in rich YP medium (1% yeast extract, 2% Bacto peptone, Sigma-Aldrich), synthetic complete (YNB) medium (0.67% yeast nitrogen base without amino acids, supplemented with 1.92 g/L of yeast synthetic drop-out media without uracil, Sigma-Aldrich) or optimized [48] YNB medium (O-YNB, containing 1.34% yeast nitrogen base without amino acids, supplemented with 3.84 g/L of yeast synthetic drop-out media without uracil and 1.5% casaminoacids, Sigma-Aldrich), with 2% maltose, glucose or xylose as carbon source. When required, 2% Bacto agar, 0.5 g/L zeocin (Invivogen, San Diego, CA, USA) or 0.2 g/L geneticin sulfate (G-418, Sigma-Aldrich) were added to the medium. The pH of the medium was adjusted to pH 5.0 with HCl, or to pH 8.0 (adjusted with NaOH) when zeocin was used. Cells were pre-grown in YNB-2% maltose and used to inoculate new medium containing the carbon sources tested with an initial cell density of 0.1 optical density at 600 nm (*A*_600nm_) measured with a Cary 60 UV-VIS spectrophotometer (Agilent Technologies, Santa Clara, CA, USA). Growth was performed aerobically in cotton-plugged Erlenmeyer flasks filled to 1/5 of the volume with medium at 28 °C with 160 rpm orbital shaking. Cellular growth was followed by absorbance measurements at 600 nm (*A*_600nm_), and culture samples were harvested regularly, centrifuged (5000× *g*, 1 min at 4 °C), and supernatants were used for the quantification of substrates and fermentation products, as described below. Alternatively, yeast strains were grown in 100 μL of O-YNB medium (lacking uracil) containing 2% maltose, glucose or xylose at 30 °C in 96-well plates in a Tecan INFINITY M200 PRO microplate reader (Tecan Austria GMBH, Grodig, Austria). All wells in the plate were tightly sealed with AccuClear Sealing Film for qPCR (E & K Scientific, Santa Clara, CA, USA), and thus this growth condition should be considered oxygen limited (or microaerobic), as no significant growth is observed when fully respiratory substrates (e.g., ethanol or glycerol) are used. The growth of each culture was monitored by measuring the *A*_600nm_ every 10 h, with high-intensity orbital shaking between measurements. After ~190 h of growth, the plates were centrifuged (3500× *g*, 10 min at 4 °C) and the supernatants used for ethanol determination, as described below. For batch fermentations, cells were pre-grown in synthetic complete YNB medium containing 2% maltose for 20 h at 28 °C; the cells were collected by centrifugation at 6000× *g* for 5 min at 4 °C and washed twice with sterile water and inoculated at a high cell density (10.0 ± 0.5 g dry cell weight (DCW)/L) into 25 mL of synthetic YNB medium containing 2% glucose or xylose. Batch fermentations under oxygen-limited or microaerobic conditions were performed at 30 °C in closed 50-mL bottles with a magnetic stir bar to allow mild agitation (100 rpm). Samples were collected regularly and processed as described above.

### 2.2. Molecular Biology Techniques

Standard methods for DNA manipulation and analysis, as well as bacterial and yeast transformation, were employed [52,53]. The genomic DNA from the *Sp. passalidarum* and *Sp. arborariae* strains was purified using a YeaStar Genomic DNA kit (Zymo Research, Irvine, CA, USA). Based on the genome sequence of *Sp. arborariae* and *Sp. passalidarum* [38,40], primers were designed (Table 1) to amplify genes with high sequence similarity to the *XUT1* gene from *Sc. stipitis* (*SsXUT1* [27]), introducing restriction sites for cloning into multicopy shuttle vectors containing strong and constitutive promoters and terminators (pPGK and p423-GPD, Table 1), as well as the *URA3* gene used as selective marker. Alternatively, the *SpXUT1* gene was amplified using a reverse primer (pPGK-*SpXUT1*ΔC-R) that introduces a premature stop codon and removes the last 22 amino acid residues of the protein. In the case of the *SaXUT1* gene, the permease was truncated both in the N- and C-terminal domains, removing the first 17 amino acid residues (primer pGPD-SaXUT1ΔNC-F, which introduces methionine after the 17 deleted residues) and removing the last 18 amino acid residues (primer pGPD-SaXUT1ΔNC-R) as described for the SpXUT1 permease.

In another approach, we knocked out two α–arrestin genes, the *ROD1* and/or *ROG3*, known to be involved in sugar transporter endocytosis, in the strain DLG-K1 using the PCR-based gene replacement procedure [49]. Briefly, the LoxP-*KanMX6-*LoxP knockout cassette from plasmid pUG6 (Table 1) was amplified with primers ROD1Δ-F and ROD1Δ-R (Table 1), and the resulting PCR product of 1615-bp (flanked by ~40 bp of upstream and downstream regions of the *ROD1* gene) containing the *KanMX6* gene was used to transform competent yeast cells. After 2-h cultivation in YP-2% glucose, the cells were plated on the same medium containing G-418 and incubated at 28 °C. G-418-resistant isolates were tested for proper genomic integration of the LoxP-*KanMX6*-LoxP cassette at the *ROD1* locus by diagnostic colony PCR using 4 primers (V-ROD1-F, V-ROD1-INT-F, V-ROD1-R and V-Kan^R^-F, Table 1). This set of primers amplified a 3556-bp fragment (primers V-ROD1-F and V-ROD1-R) or a 2633-bp fragment (primers V-ROD1-INT-F and V-ROD1-R) from a normal *ROD1* locus, or yielded a 2720-bp fragment (primers V-ROD1-F and V-ROD1-R) and a 1500-bp fragment (primers V-Kan^R^-F and V-ROD1-R) if the LoxP-*KanMX6-*LoxP cassette was correctly integrated at the *ROD1* locus, producing strain DLG-K1∆R1 (*rod1*∆::LoxP-*KanMX6*-LoxP, Table 1). A similar approach was used to delete the *ROG3* gene from the strains DLG-K1 or DLG-K1ΔR1. The LoxP-*Ble*^R^-LoxP knockout cassette from plasmid pUG66 (Table 1) was amplified with primers ROG3Δ-F and ROG3Δ-R (Table 1), and the resulting PCR product of 1265-bp (flanked by ~40 bp of the upstream and downstream regions of the *ROG3* locus) containing the *Ble*^R^ gene was used to transform competent cells. After 2-h cultivation in YP-2% glucose, the cells were plated on the same medium containing zeocin and incubated at 28 °C. Zeocin-resistant isolates were tested for proper genomic integration of the LoxP-*Ble*^R^-LoxP cassette at the *ROG3* locus by diagnostic colony PCR using 4 primers (V-ROG3-F, V-ROG3-INT-F, V-ROG3-R and V-Ble^R^-F, Table 1). This set of primers amplified a 2967-bp fragment (primers V-ROG3-F and V-ROG3-R) or a 1249-bp fragment (primers V-ROG3-INT-F and V-ROG3-R) from a normal *ROG3* locus, or yielded a 1757-bp fragment (primers V-ROG3-F and V-ROG3-R) and a 750–bp fragment (primers V-Ble^R^-F and V-PHO13-R) if the *LoxP*-*Ble*^R^-*LoxP* module replaced and deleted the *ROG3* gene, producing strain DLG-K1ΔR3 (*rog3*Δ::LoxP-*Ble*^R^-LoxP, Table 1) and strain DLG-K1∆R1ΔR3 (*rod1*∆::LoxP-*KanMX6*-LoxP and *rog3*Δ::LoxP-*Ble*^R^-LoxP, Table 1).

### 2.3. Transport Assays

The DLG-K1 yeast strain transformed with plasmids pPGK-*SsXUT1* or pPGK-*SpXUT1*, which were grown in YNB-2% maltose to mid-log phase (*A*_600nm_ of 0.6–1.0), were centrifuged, washed twice with cold distilled water and suspended in water to a cell density of 20 g DCW/L. The uptake of D-[U-^14^C]glucose or D-[U-^14^C]xylose (both from Amersham, Little Chalfont, UK) was determined by placing 20 μL of the yeast suspension with 20 μL of 100 mM Tris-citrate buffer, pH 5.0, in the bottom of 8-mL Röhren tubes (Sarstedt AG & Co. KG, Numbrecht, Germany). The tube was incubated at 25 °C for 5 min, and the reaction started by adding 10 μL of the radiolabeled substrate (102–104 cpm/nmol) at the desired final sugar concentration with vigorous shaking. After 5 s, the reaction was stopped with 5 mL of ice-cold distilled water and vigorous shaking, immediately filtered in Whatman glass microfiber GF/C membranes (2.4 cm diameter), and the filters were washed twice with 10 mL of ice-cold distilled water. The filters were placed into scintillation vials containing 6 mL of liquid scintillation cocktail (OptiPhase ‘HiSafe’ 2, Wallac, Turku, Finland), and the radioactivity retained on filters was counted using a liquid scintillation counter (Tri-Carb^TM^ 1600 CA, Packard, Downers Grove, IL, USA). The kinetic parameters (*K*_m_ and *V*_max_) of glucose and xylose transport by the cloned permeases were determined by fitting the experimental data to the Michaelis–Menten equation, using SigmaPlot v. 11.0 (Systat Software Inc., San Jose, CA, USA).

### 2.4. Analytical Methods

Glucose, xylose, xylitol, glycerol and ethanol were separated and quantified by high-performance liquid chromatography (Prominence HPLC system) equipped with a RID-20A refractive index detector (Shimadzu Co., Tokyo, Japan) using an Aminex HPX-87H column (Bio-Rad Laboratories, Hercules, CA, USA). The HPLC apparatus was operated at 50 °C using 5 mM H_2_SO_4_ as mobile phase at a flow rate of 0.6 mL/min and 10 μL injection volume.

### 2.5. Prediction of Lysine Residues with Ubiquitinylation Potential

The amino acid sequences of the transporters were retrieved from NCBI [54] in FASTA format and aligned using Clustal Omega [55]. The transmembrane α-helices were predicted with the PRALINE program [56,57] using the PSIPRED secondary structure prediction and the HMMTOP transmembrane structure prediction options. The lysine residues with ubiquitinylation potential in the permease sequences were determined with the UbPred program [58,59].

## 3. Results

The SsXUT1 permease from *Sc. stipitis* has been previously studied in *S. cerevisiae*, showing the highest preference for xylose among several other transporters analyzed [24,27]. Using the amino acid sequence of this transporter as query in a basic local alignment search tool (BLAST), two other putative permeases from the xylose-fermenting yeasts *Sp. passalidarum* and *Sp. arborariae* were identified. The open-reading frame (ORF) from the *Sp. passalidarum* genome encodes a protein of 576 amino acids with 78.8% sequence identity with SsXUT1, while the ORF present in the genome of *Sp. arborariae* encodes a protein with 572 amino acids and 79.5% sequence identity with SsXUT1. The genes were cloned into multicopy plasmids, as described in the Materials and Methods section, and named *SpXUT1* and *SaXUT1*, respectively. The characterization of these putative transporters relied on the transformation of the DLG-K1 (*hxt*-null) strain with those plasmids. The resulting strains were pre-grown in YNB-2% maltose and inoculated (initial *A*_600nm_ of 0.1) into YNB medium containing 2% glucose or xylose. However, under those conditions, no significant growth or sugar consumption was observed after 120 h of incubation. Figure 2 shows, as example, the data for strain DLG-K1 transformed with plasmid pPGK-*SpXUT1*. An absence of growth was also observed when the cells were inoculated in media containing 2% fructose or galactose.

However, after growth on 2% maltose, the strain expressing the SpXUT1 permease takes up ^14^C-glucose and ^14^C-xylose in a similar manner to the one expressing the SsXUT1 permease (Figure 3). In fact, the SsXUT1 permease allows ^14^C-glucose transport with a *K*_m_ of 24.5 mM, but with a low *V*_max_, while ^14^C-xylose transport occurs with a much lower affinity, although with higher capacity (Figure 3, Table 2). In the case of the SpXUT1 permease ^14^C-glucose, transport is mediated with a similar affinity of 26.1 mM and also a low *V*_max_, and ^14^C-xylose transport is also mediated with an even lower affinity but high capacity when compared to glucose uptake (see Table 2).

Despite the high sequence identity between these three transporters, a significant difference was observed in their N- and C-terminal cytoplasmic domains regarding the presence of lysine residues with high ubiquitinylation potential (Figure 4). SsXUT1 permease does not have any lysine in its N-terminal domain and has a single one with high ubiquitinylation potential at the end of the C-terminal domain (residue K-558). SpXUT1 permease has the highest number of lysine residues at their cytoplasmic domains, one with medium ubiquitinylation potential in its N-terminal domain (residue K-17) and three with high ubiquitinylation potential in their C-terminal cytoplasmic domain (residues K-555, K-557 and K-567, see Figure 4). In the case of the SaXUT1 permease, the lysine residues with ubiquitinylation potential were the same as the ones identified for SpXUT1, although residues K-555 and K-567 were considered as being of medium ubiquitinylation potential by the UbPred program [58].

Considering that such lysine residues with ubiquitinylation potential could be involved in removing the transporters from the plasma membrane through endocytosis [60], we decided to remove these terminal lysine residues by simply truncating the permeases. In the case of the SpXUT1 permease, we removed the last 22 amino acid residues of the protein by introducing a premature stop codon during cloning, as described in Materials and Methods, and this modified transporter was denominated SpXUT1ΔC. For the SaXUT1 permease, we not only removed the last 18 amino acid residues, but also the first 17 amino acid residues, producing the SaXUT1ΔNΔC permease lacking all lysine residues with ubiquitinylation potential both from the N- and C-terminal domains.

As can be seen in Figure 2, when the DLG-K1 strain was transformed with the pPGK-*SpXUT1*ΔC plasmid, the cells were able to efficiently grow on, consume and ferment glucose (Figure 2a), while xylose consumption also occurred, although being delayed and incomplete (Figure 2b). Nevertheless, the strain expressing the SpXUT1ΔC permease was able to ferment xylose, producing both ethanol and xylitol, while the strain expressing the full-length transporter (SpXUT1) was unable to metabolize the pentose as described above (Figure 5). Indeed, the strain expressing the SpXUT1ΔC permease was more efficient in xylose fermentation (higher rates of xylose consumption, as well as higher production of ethanol and xylitol) than the previously characterized strain expressing the SsXUT1 transporter [27]. In the case of the SaXUT1 permease, we also observed clear growth improvement in glucose or xylose medium when both the N- and C-terminal domains were truncated (SaXUT1ΔNΔC), but not with the impressive results obtained with the SpXUT1ΔC permease described above. For example, under the microaerobic conditions shown in Figure 5, the strain expressing SaXUT1ΔNΔC only consumed ~5 g/L of xylose and produced ~1.5 g/L of ethanol after 48 h, while the strain expressing the full length SaXUT1 permease was unable to consume the pentose (Appendix A).

Since our results suggested that the ubiquitinylation of heterologous monosaccharide permeases impairs their successful expression in *S. cerevisiae*, and since this post-translational modification that triggers the endocytosis of the plasma membrane transporters depends on the activity of α–arrestins that function as adaptors for the E3 ubiquitin ligase encoded by the essential gene *RSP5* [61,62], we investigated the impact of the knockout of α–arrestins on the functional expression of the cloned transporters in *S. cerevisiae*. Although this yeast has 14 known α–arrestins, we focused on two α–arrestins known to be involved in sugar transporter endocytosis: i) ROD1 (also known as ART4 for *a*rrestin*-r*elated *t*rafficking adaptors), which has been shown to mediate the ubiquitinylation and endocytosis of the high-affinity HXT6 and HXT4 glucose permeases [63,64,65], the low-affinity HXT1 and HXT3 glucose permeases [66], as well as GAL2, the galactose permease [67]; and ii) its paralog ROG3 (also known as ART7), which has been shown to be implicated in the ubiquitinylation and endocytosis of the low-affinity HXT1 and HXT3 glucose permeases [65,66].

To verify the influence of these two α–arrestins on the functional expression of the SpXUT1 and SaXUT1 permeases in *S. cerevisiae*, we constructed, in our *hxt*-null DLG-K1 chassis (Table 1), knockout mutants in *ROD1* (DLG-K1∆R1), *ROG3* (DLG-K1∆R3) or both *ROD1* and *ROG3* (DLG-K1∆R1∆R3). Figure 6 shows the growth patterns of these three strains, transformed with plasmids pGPD-*SaXUT1* or pPGK-*SpXUT1*, in three carbon sources: maltose, glucose and xylose. Analyzing the data, it is evident that the deletion of both α-arrestins (*rod1*Δ, *rog3*Δ) has a negative impact on growth, even with the control carbon source (2% maltose). While DLG-K1 and single α-arrestin mutants produced 8–9 g ethanol/L (Table 3), strain DLG-K1∆R1∆R3 transformed with both plasmids produced less ethanol (6–8 g/L) from 2% maltose. Knockout of both genes not only did not improve the results when the cells were grown in the other two carbon sources, but in fact it had a negative impact on the performance of the *hxt*-null strains expressing either of the XUT1 permeases (Figure 6).

Growth under microaerobic conditions in glucose was clearly improved in the *rog3*Δ strain DLG-K1ΔR3 expressing either of the XUT1 permeases (Figure 6). The *rog3*Δ cells expressing the SpXUT1 transporter produced 7.1 ± 0.2 g ethanol/L in 2% glucose, while the expression of SaXUT1 permease produced only 1.1 ± 0.1 g ethanol/L (Table 3), a lower performance, as already mentioned, for strains expressing this permease. However, *ROD1* and/or *ROG3* knockout did not improve the utilization of xylose in the cells expressing the SpXUT1 permease, suggesting that other α–arrestins might be involved in ubiquitinylation of this transporter in the presence of the pentose. A different pattern was obtained with the strains expressing the SaXUT1 permease, as the DLG-K1ΔR3 strain showed some growth in xylose medium (Figure 6) but without ethanol production. The improvements observed with the α–arrestins knockout strains shown in Figure 6 (performed in a microplate reader with 96-well plates) were confirmed when these strains (expressing both truncated transporters) were grown aerobically in cotton-plugged Erlenmeyer flasks with 2% glucose (Appendix A).

## 4. Discussion

Our group previously used a *S. cerevisiae* chassis strain (DLG-K1) to clone and characterize three sugar transporters from *Sc. stipitis* that allowed growth and fermentation in xylose medium [27]. One of the transporters (SsXUT1) was already known to allow growth in xylose [24], but the other two (SsHXT2.6 and SsQUP2) were completely new and even annotated for other functions in the published genome sequence of *Sc. stipitis* [27]. Interestingly, these three permeases have only one lysine residue with high ubiquitinylation potential, either at the C-terminal cytoplasmic domain (SsXUT1 and SsQUP2) or at the N-terminal domain (SsHXT2.6) (see SsXUT1 in Figure 4). In contrast, the two putative sugar transporters cloned in this work, originating from the xylose-fermenting yeasts *Sp. passalidarum* and *Sp. arborariae* and showing high amino acid sequence identity to SsXUT1, had several lysine residues with high ubiquitinylation potential at their C-terminal domain, as well as lysine residues with medium ubiquitinylation potential, also at their N-terminal cytoplasmic domain (see Figure 4).

In *S. cerevisiae*, sugar transporters can be ubiquitinylated and endocytosed depending not only on the type (and quantity) of the sugar present in the medium, but also in response to other environmental perturbations [60,61,62]. For example, low-affinity and high-capacity glucose transporters (e.g., *HXT1* and *HXT3*) are endocytosed in the absence of glucose (or in the presence of other carbon sources, such as galactose, ethanol or lactate), while high-affinity glucose transporters (e.g., *HXT2*, *HXT6* and *HXT7*) are removed from the plasma membrane in the presence of high glucose concentrations, but also in the absence of glucose (or in the presence of other carbon sources) [41,64,66,67,68,69]. In turn, the knowledge about ubiquitinylation and endocytosis in the expression of heterologous monosaccharide transporters in *S. cerevisiae* is still limited [31].

Indeed, several efforts to express heterologous monosaccharide transporters in *S. cerevisiae* have been unsuccessful. For example, Young et al. [24] cloned 23 transporters from different organisms (bacteria, plant and several yeasts) and expressed them in an *hxt*-null strain with a xylose utilization pathway. Of these 23 heterologous transporters, only 7 (including SsXUT1) conferred significant growth phenotypes on one or more of six different carbon sources tested. Several other recent publications, reporting attempts at the identification and cloning of monosaccharide transporters from yeasts and fungi, reveal low success rates [70,71,72]. Among the many possible reasons for the unsuccessful expression of a putative sugar transporter in *S. cerevisiae* are protein misfolding, problems with protein trafficking to the plasma membrane, incompatible lipid environment of the host, as well as the stability at the cell surface, including transporter ubiquitinylation and endocytosis.

Ubiquitinylation of lysine residues at the N- or C-terminal cytoplasmic domains is part of the molecular signals used in the turnover of sugar permeases, and several studies have shown that the modification or removal of such residues can stabilize the proteins in the plasma membrane. For example, the *S. cerevisiae HXT1* transporter has 4 lysine residues at its N-terminal domain that are involved in its ubiquitinylation and endocytosis, and by truncating the whole N-terminal domain (deletion of 56 residues to remove all 4 lysine residues), the stability of the transporter in the plasma membrane is improved [68]. We have recently used this strategy to enhance xylose consumption and fermentation in an industrial *S. cerevisiae* strain [73], while site-directed mutation of these four residues in the *HXT1* transporter was also shown to improve growth in this carbon source [74]. Another example involves one of the few yeast active xylose transporters characterized in *S. cerevisiae*, the *Candida intermedia* glucose/xylose H^+^-symporter, CiGXS1 [75]. This transporter was cloned in an *hxt*-null *S. cerevisiae* strain and submitted to several rounds of mutagenesis aiming at improving glucose-xylose co-fermentation, and among several interesting mutations found, there were some that truncated the C-terminal domain of the permease [76]. A systematic analysis of this C-terminal domain revealed that the best results in terms of growth in glucose/xylose medium occurred when 27 amino acids were deleted from the C-terminus of the CiGXS1 permease, which included two lysine residues [76]. Finally, C-terminal truncations that enhance the stabilization and transport activity of two heterologous cellobiose transporters expressed in *S. cerevisiae* have also been reported, eliminating 1-2 lysine residues present at their C-terminal domains probably involved in ubiquitinylation and endocytosis [48]. Thus, our approach of truncating the SpXUT1 or SaXUT1 permeases, removing the potentially ubiquitinylate lysine residues at their N- and/or C-terminal domains, is certainly an interesting strategy to increase the repertory of heterologous sugar (especially pentose) transporters in genetic engineering approaches of *S. cerevisiae* strains for the efficient conversion of sugar into biofuels and biochemicals. It would be interesting to verify how other combinations of removing the lysine residues by truncating the permeases, for example, deleting the N-terminal domain of the SpXUT1 permease, or only the C-terminal domain of the SaXUT1 transporter, impact the functionality of the cloned permeases. In another approach, the different lysine residues identified in the permeases (see Figure 4) could be removed by site-directed mutagenesis (changing them to alanine or arginine residues, e.g., [48,68,74]), separately or in different combinations, to systematically identify which lysine residues are involved in the covalent attachment of ubiquitin and thus endocytosis of the permeases.

In addition to eliminating lysine residues from the transporters by truncating their cytoplasmic terminal domains, we also tested the involvement of two α–arrestins (*ROD1* and *ROG3*) known to be involved in sugar transporter ubiquitinylation and endocytosis in *S. cerevisiae*. While the knockout of *ROD1* or both α–arrestins in our *hxt*-null strain had no effect on the growth of the strains expressing the two XUT1 permeases, the knockout of *ROG3* had a positive effect with both permeases during growth in glucose. In xylose medium, only the SaXUT1-expressing strain DLG-K1ΔR3 had its growth enhanced (Figure 6). This more complex situation might reflect the fact that most α–arrestins have overlapping functions when promoting the ubiquitinylation of the permeases by the *RSP5* ubiquitin ligase, and many transporters are regulated by two or even up to four α–arrestins, reflecting the diverse signaling pathways that mediate sugar transporter ubiquitinylation and endocytosis triggered by changing environmental conditions [48,61,62,65,66,67]. Nevertheless, our results highlight the involvement of the *ROG3* α–arrestin in the downregulation of heterologous SpXUT1 and SaXUT1 permeases expressed in *S. cerevisiae* in the presence of glucose, while a more complex situation is evident when xylose is the carbon source used. It would be interesting to further explore the involvement of these and other α–arrestins in the efficient heterologous expression of permeases in *S. cerevisiae*, possibly in combination with mutations in the lysine residues as referred to above.

## 5. Conclusions

In the present work, we have cloned two XUT1 permeases from the xylose-fermenting yeasts *Sp. passalidarum* and *Sp. arborariae*. While these two transporters seemed to be not functional in *S. cerevisiae*, the truncation of the N- and/or C-terminal domains, eliminating lysine residues with ubiquitinylation potential, allowed the functional expression of the transporters in an *hxt*-null strain. Thus, this might be an interesting strategy to increase the repertory of heterologous sugar transporters in the engineering of *S. cerevisiae* for improved sugar utilization or simply to characterize putative sugar transporter from different eukaryotic origins. This work, therefore, highlights the importance of post-translational modifications in the correct expression of novel sugar transporters in recombinant *S. cerevisiae* strains.

## Figures and Tables

**Figure 1 jof-08-00084-f001:**
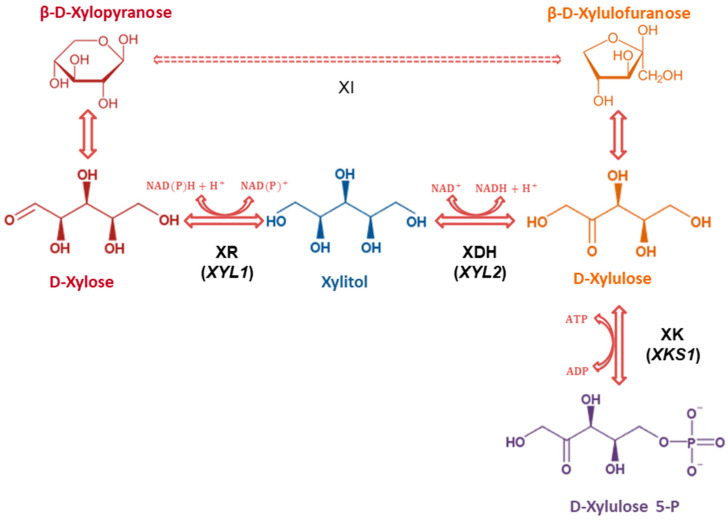
Xylose utilization pathways in fungi. Xylose can be directly isomerized (XI) to xylulose, or this pentose is first reduced (XR) to xylitol and then oxidized (XDH) to xylulose. Both pathways require the activity of xylulokinase (XK) that will enable xylulose-5-P to enter the nonoxidative part of the pentose phosphate pathway and glycolysis [12]. The overexpressed genes in strain DLG-K1 are indicated.

**Figure 2 jof-08-00084-f002:**
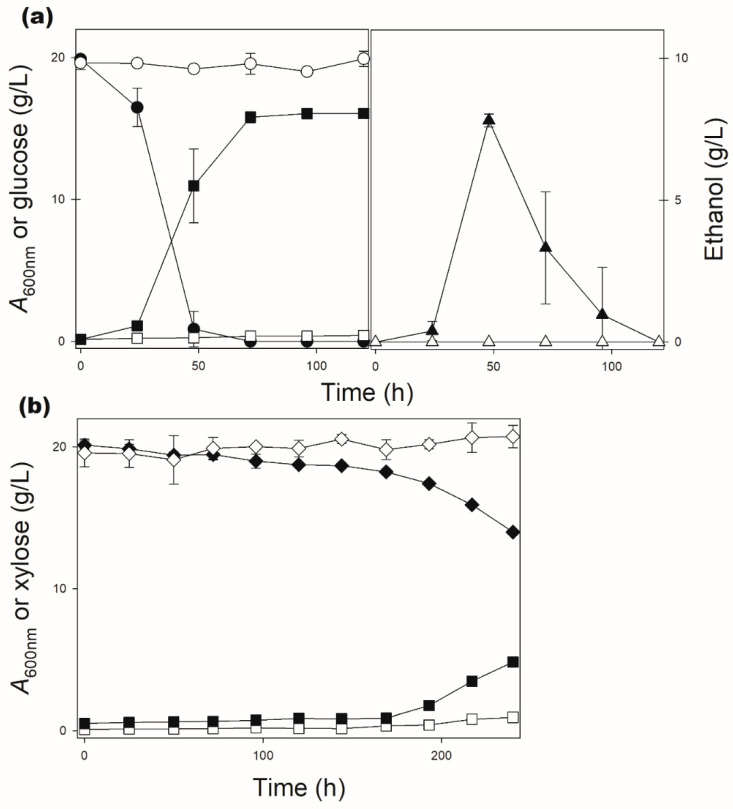
Aerobic growth of strain DLG-K1 transformed with plasmid pPGK-*SpXUT1* (open symbols) or plasmid pPGK-*SpXUT1*ΔC (black symbols) in YNB medium (lacking uracil) containing 2% glucose (**a**) or xylose (**b**) as carbon source, at 30 °C. At the indicated time points, the cell growth (squares), glucose (circles), xylose (diamonds) and ethanol (triangles) concentrations were determined. No ethanol was produced during xylose consumption.

**Figure 3 jof-08-00084-f003:**
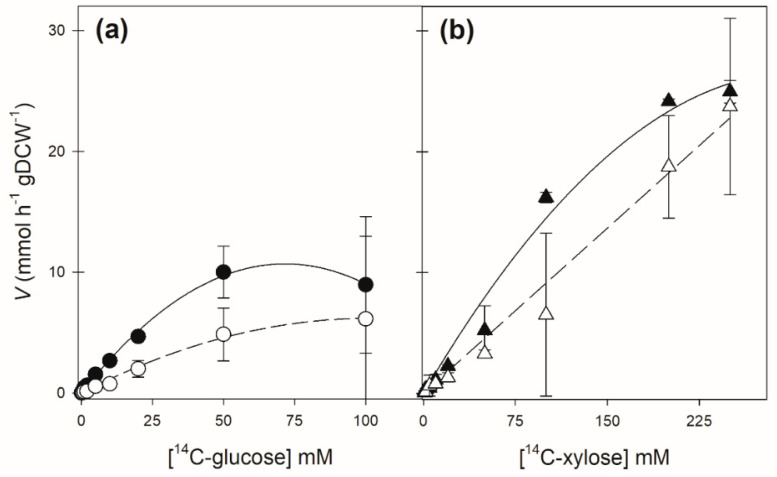
Kinetics of glucose (**a**) or xylose (**b**) transport by strain DLG-K1 transformed with plasmid pPGK-*SsXUT1* (black symbols) or plasmid pPGK-*SpXUT1* (open symbols). Cells pre-grown in 2% maltose were used to determine the initial rates of uptake of the indicated labeled sugar concentrations at 25 °C.

**Figure 4 jof-08-00084-f004:**
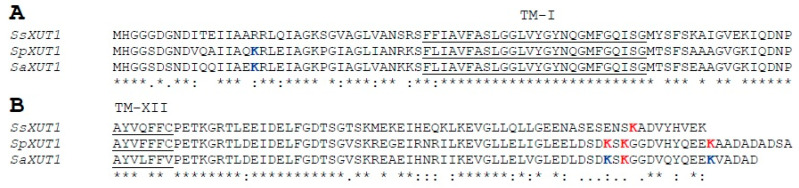
Sequence alignment of the N-terminal (**A**) and C-terminal (**B**) cytoplasmic domains of the SsXUT1, SpXUT1 and SaXUT1 transporters. The protein sequences were aligned using Clustal Omega [55]; the transmembrane α-helices (underlined residues) were predicted with the PRALINE program [56], and the lysine residues with medium (blue) or high (red) ubiquitinylation potential were determined with the UbPred program [58].

**Figure 5 jof-08-00084-f005:**
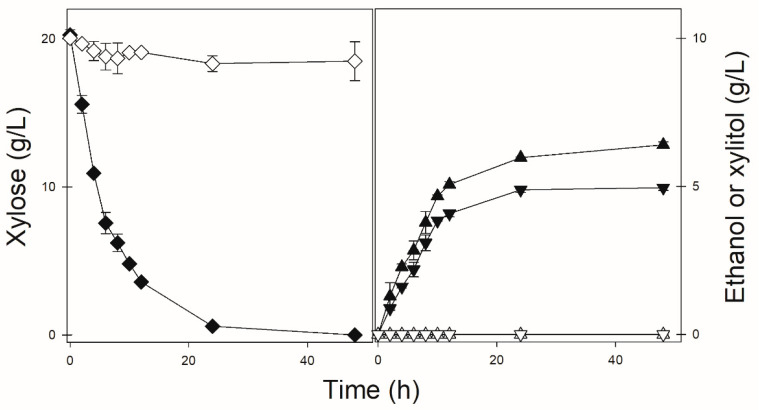
Xylose fermentation under microaerobic conditions by the DLG-K1 strain transformed with the pPGK-*SpXUT1* (open symbols) or pPGK-*SpXUT1*ΔC (black symbols) plasmids. The batch fermentation was performed with high cell concentrations (10 g DCW/L), at 30 °C, and the amount of xylose (diamonds), ethanol (tringles) and xylitol (inverted triangles) in the medium was determined at the indicated time points.

**Figure 6 jof-08-00084-f006:**
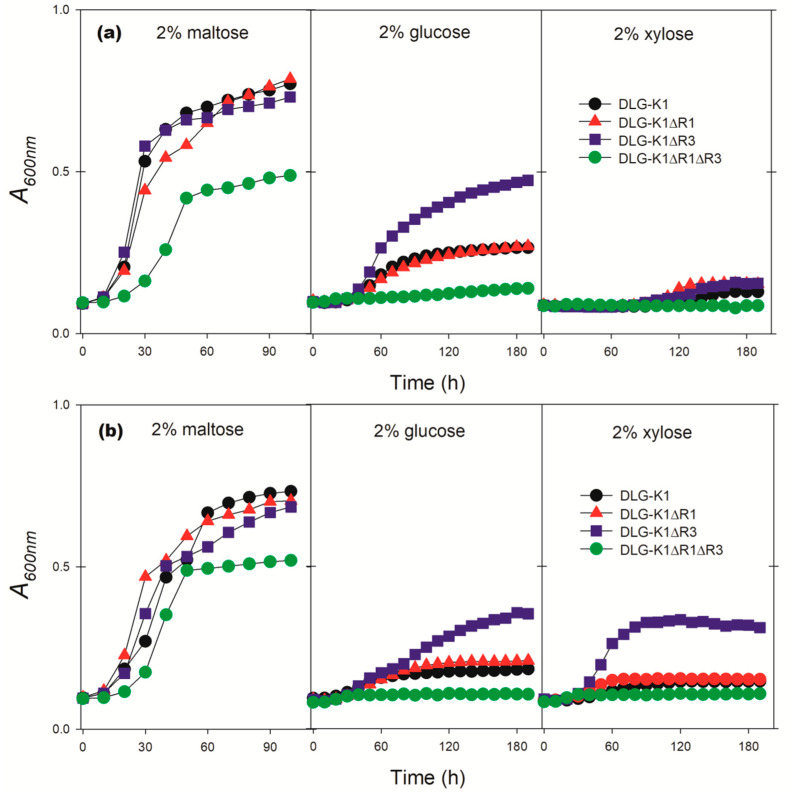
Growth under microaerobic conditions of the indicated *hxt*-null strains containing the pPGK-*SpXUT1* (**a**) or pGPD-*SaXUT1* (**b**) plasmids on O-YNB medium containing 2% of the indicated carbon sources in a microplate reader at 30 °C.

**Table 1 jof-08-00084-t001:** Yeast strains, plasmids and primers used in this study.

Strains, Plasmids and Primers	Relevant Features, Genotype or Sequence	Source
Yeast strains:		
*Sp. arborariae* UFMG-HM19.1A^T^	Isolated from rotting wood in Minas Gerais, Brazil	[39]
*Sp. passalidarum* UFMG-CM-Y474	Isolated from rotting wood in Roraima, Brazil	[17]
*S. cerevisiae* DLG-K1	*MAT**a** hxt1*Δ::*HIS3*::Δ*hxt4 hxt2*Δ::*HIS3 hxt5*::*LEU2 hxt7*::*HIS3 hxt3*Δ::*LEU2*::*hxt6 gal2*Δ::DR ^1^ *ura3-52 his3-11,15 leu2-3,112 MAL2 SUC2**AUR1*::pAUR-XKXDHXR	[21]
*S. cerevisiae* DLG-K1∆R1	Isogenic to DLG-K1, but *rod1*∆::LoxP-*KanMX6*-LoxP	This work
*S. cerevisiae* DLG-K1∆R3	Isogenic to DLG-K1, but *rog3*∆::LoxP-*Ble*^R^-LoxP	This work
*S. cerevisiae* DLG-K1∆R1∆R3	Isogenic to DLG-K1, but *rog3*∆::LoxP-*Ble*^R^-LoxP *rod1*∆::LoxP-*KanMX6*-LoxP	This work
Plasmids:		
pUG6	LoxP-P_TEF_-*KanMX6-*T_TEF_-LoxP	[49]
pUG66	LoxP-P_TEF_*-Ble^R^-*T_TEF_-LoxP	[49]
pPGK	2 µ *URA3* P_PGK1_-T_PGK1_	[50]
pGPD-426	2 µ *URA3* P_TDH3_-T_CYC1_	[51]
pPGK-*SsXUT1*	2 µ *URA3* P_PGK1_-*SsXUT1-*T_PGK1_	[27]
pPGK-*SpXUT1*	2 µ *URA3* P_PGK1_-*SpXUT1-*T_PGK1_	This work
pPGK-*SpXUT1*ΔC	2 µ *URA3* P_PGK1_-*SpXUT1*ΔC-T_PGK1_	This work
pGPD-*SaXUT1*	2 µ *URA3* P_TDH3_-*SaXUT1*-T_CYC1_	This work
pGPD-*SaXUT1*ΔNC	2µ *URA3* P_TDH3_-*SaXUT1*ΔNC-T_CYC1_	This work
Primers: ^2^		
pPGK-SpXUT1-F	AGATCG**GAATTC**AAGCTTATGCACGGAGGTTCAGACG	This work
pPGK-SpXUT1-R	GCC**GGATCC**GGCTTAAGCACTGTCAGCATCAGC	This work
pPGK-SpXUT1ΔC-R	GGC**GGATCC**AAATTAGTCAGAGTCTAATTCTTCTCCGCC	This work
pGPD-SaXUT1-F	AGATCGGAATTCAAGCTT**GGATCC**ATGCACGGAGGTTCAGATAGTAA	This work
pGPD-SaXUT1-R	GCC**CTCGAG**GTCGACCCCGGGGGCTTAATCAGCATCAGCAACCTTTTC	This work
pGPD-SaXUT1ΔNC-F	GCC**GGATCC**AAAATGCGTTTAGAAATCGCCGGTAAACC	This work
pGPD-SaXUT1ΔNC-R	GCC**CTCGAG**GTCGACTTAATCAGAATCTAAGTCTTCTAATCC	This work
ROD1Δ-F	*ATGTTTTCATCATCATCTCGACCTTCAAAAGAGCCATTAC* CCAGCTGAAGCTTCGTACGC	This work
ROD1Δ-R	*CTATGAGCGATCCCGTTTTGTGAACATCTCCATTAAATTA* GCATAGGCCACTAGTGGATC	This work
ROG3Δ-F	*GGCGTTGATAAAGAGCCAATATCTATTGTTGCTACATAGA* CCAGCTGAAGCTTCGTACGC	This work
ROG3Δ-R	*CGACTATCGTTTGTTACCCTTTGATAGAAAACCTCCCATA* GCATAGGCCACTAGTGGATC	This work
V-ROD1-F	AGTCGAGTCCCTTGGTACAT	This work
V-ROD1-INT-F	CTGCCGTCACTTATGCTCTG	This work
V-ROD1-R	CGAATGATGTCTGTGGGATC	This work
V-ROG3-F	GCAAGTACAGAGTCCTACCA	This work
V-ROG3-INT-F	CTGTGTGCAAGATTGTGATG	This work
V-ROG3-R	GCCAGTTAGAGTGCGTAAAT	This work
V-Kan^R^-F	CCGGTTGCATTCGATTCC	This work
V-Ble^R^-F	CCTTCTATGAAAGGTTGGGC	This work

^1^ Direct repeat; ^2^ Bold sequences indicate restriction enzyme sites (*Bam*HI*, Eco*RI or *Xho*I) used for cloning; underlined sequences allow amplification of genes or the transformation modules present in plasmids pUG6 and pUG66, and italicized sequences are homologous to the upstream and downstream region of the target genes that were deleted.

**Table 2 jof-08-00084-t002:** Kinetic parameters of ^14^C-glucose and ^14^C-xylose transport by the SsXUT1 and SpXUT1 permeases at 25 °C, in cells grown in YNB-2% maltose.

Strain	Transport of ^14^C-Glucose		Transport of ^14^C-Xylose	
	*K*_m_ (mM)	*V*_max_ (mmol h^−1^ gDCW^−1^)	*K*_m_ (mM)	*V*_max_ (mmol h^−1^ gDCW^−1^)
DLG-K1 + pPGK-*SsXUT1*	24.5 ± 4.1	10.8 ± 1.8	417.7 ± 176	72.4 ± 27
DLG-K1 + pPGK-*SpXUT1*	26.1 ± 11.5	4.3 ± 1.9	711 ± 550	72.3 ± 7.5

**Table 3 jof-08-00084-t003:** Ethanol production under microaerobic conditions by strains expressing the SpXUT1 or SaXUT1 permeases, determined at the end of growth on O-YNB medium containing the indicated carbon sources in a microplate reader at 30 °C.

Plasmid and Carbon Source	Ethanol Produced (g/L) by Strains:			
	DLG-K1	DLG-K1∆R1	DLG-K1∆R3	DLG-K1∆R1R3
pPGK-*SpXUT1*				
2% maltose	9.0 ± 0.2	9.3 ± 0.6	9.3 ± 0.5	7.6 ± 0,2
2% glucose	0.0	0.0	7.1 ± 0.2	0.0
pGPD-*SaXUT1*				
2% maltose	8.3 ± 0.1	8.2 ± 0.1	7.9 ± 0.3	5.8 ± 0.1
2% glucose	0.0	0.0	1.1 ± 0.1	0.0

## Data Availability

Not applicable. All data is contained within the article.

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
