# Peer review of "Strategies for Efficient Expression of Heterologous Monosaccharide Transporters in Saccharomyces cerevisiae"

_jof, 2022, doi:10.3390/jof8010084_

Round 1

Reviewer 1 Report

The manuscript reports the characterization of two xylose transporters heterologously expressed in S. cerevisiae. By truncating the lysine-containing terminal domains or deleting arrestin genes related to sugar transporter ubiquitinylation and endocytosis, strains expressing these transporters showed better performance in xylose utilization. My major concern is the novelty of this work, although its title is “new strategies for …”. The manuscript highlights the strategy of eliminating lysine residues from the cytoplasmic domains of the transporters by truncating their cytoplasmic terminal domains, but this strategy is not new, and has been proven efficient in previous studies including their own [58, 63].

Other comments include:

  1. The role of the lysine residues in the C- and N-termini of the permeases might be better discussed if SsXUT1ΔN, SpXUT1ΔN, SpXUT1ΔNΔC, SpXUT1ΔN and SaXUT1ΔC could also be tested.
  2. Although the variation in ethanol production upon deletion of rog3 and/or rod1 was mentioned in the text, it was not shown in Figure 5. Moreover, the results for shaking-flask cultures are not shown. Presentation of these data would be helpful.
  3. The genotype of the hxt-null strain DLG-K1 strain used in this study should be described in more details.
  4. The authors are suggested to carefully check through the manuscript and correct the grammatical errors.

Author Response

The manuscript reports the characterization of two xylose transporters heterologously expressed in S. cerevisiae. By truncating the lysine-containing terminal domains or deleting arrestin genes related to sugar transporter ubiquitinylation and endocytosis, strains expressing these transporters showed better performance in xylose utilization. My major concern is the novelty of this work, although its title is “new strategies for …”. The manuscript highlights the strategy of eliminating lysine residues from the cytoplasmic domains of the transporters by truncating their cytoplasmic terminal domains, but this strategy is not new, and has been proven efficient in previous studies including their own [58, 63].

Response to reviewer # 1.

We thanks the comments and suggestions raised by the reviewer, which we think contributed to improve the manuscript.

Regarding the “novelty” of the work, we would like to say that removing lysine residues from the cytoplasmic domains of the transporters by truncating their sequences was performed only with the endogenous (and thus totally functional) HXT1 transporter from S. cerevisiae, improving its stabilization at the plasma membrane (now references [68, 73]). As the title declares, this is a new strategy for expression of heterologous transporters which otherwise seemed to be not functional in S. cerevisiae, something that has not been described in the literature before.

Other comments include:

  1. The role of the lysine residues in the C- and N-termini of the permeases might be better discussed if SsXUT1ΔN, SpXUT1ΔN, SpXUT1ΔNΔC, SpXUT1ΔN and SaXUT1ΔC could also be tested.

We agree with the reviewer and its suggestion of truncating the transporters in several other combinations. Indeed, we were planning another approach to better define which lysine residues are responsible for the endocytosis of the permeases, since there are 3-4 different candidates at the cytoplasmic domains, by site-directed mutagenesis of each lysine separately (or in different combinations), but unfortunately we are still at the beginning of such more detailed (and scientifically interesting) approach. aggravated by the current pandemic and lockdown restrictions we are currently facing.

  1. Although the variation in ethanol production upon deletion of rog3 and/or rod1 was mentioned in the text, it was not shown in Figure 5. Moreover, the results for shaking-flask cultures are not shown. Presentation of these data would be helpful.

As explained in Materials and Methods (lines 154-161) the growth data of the strains deleted in rog3 and/or rod1 (now Figure 6) was performed with tightly sealed 96-well plates, and thus the ethanol produced could only be determined at the end of the experiment. However, in our revised version of the manuscript we have introduced a new Table 3 (top of page 12) showing the ethanol produced by the different strains after growth on maltose and glucose (no ethanol was detected when xylose was the carbon source), and the Supplementary Figure S2 shows the data obtained by the rog3 deleted strain in shaking-flask cultures with 2% glucose. From that figure is evident that truncating the cytoplasmic domain of the permease is more effective than the rog3 deletion, especially for the SpXUT1 transporter (compare with now Fig. 2), while the SaXUT1 permease could hardly allow glucose consumption by this rog3 deleted strain.

  1. The genotype of the hxt-null strain DLG-K1 strain used in this study should be described in more details.

The genotype of the DLG-K1 is now provided in more details in the Introduction (lines 93-98), and particularly in the Materials and Methods section (lines 120-134).

  1. The authors are suggested to carefully check through the manuscript and correct the grammatical errors.

The text was extensively revised and detected grammatical errors corrected.

Reviewer 2 Report

In this manuscript, the authors tested putative XUT monosaccharide transporters from xylose-fermenting species of Spathaspora for their abiility to function in Saccharomyces cerevisiae with either glucose or xylose. Interestingly, the XUTs were only functional when their endocytosis was compromised, either by deletion of lysine residues or by deletion of the ROG3 alpha-arrestin. The results highlight the importance of engineering stability of heterologous transporters in the plasma membrane to ensure their functionality for second-generation biofuels production. The manuscript thus provides both an introduction of two novel transporter candidates to the field, and also tests a strategy for their functional expression.

Minor comments:

Line 27, "growth"

Line 38, "intensified"

Line 292, "grow on"

Line 341, "either XUT1 permease"

Line 374, "endocytosed"

Line 393, "the"

Author Response

In this manuscript, the authors tested putative XUT monosaccharide transporters from xylose-fermenting species of Spathaspora for their abiility to function in Saccharomyces cerevisiae with either glucose or xylose. Interestingly, the XUTs were only functional when their endocytosis was compromised, either by deletion of lysine residues or by deletion of the ROG3 alpha-arrestin. The results highlight the importance of engineering stability of heterologous transporters in the plasma membrane to ensure their functionality for second-generation biofuels production. The manuscript thus provides both an introduction of two novel transporter candidates to the field, and also tests a strategy for their functional expression.

Response to reviewer # 2.

We thanks the comments and suggestions raised by the reviewer, which we think contributed to improve the manuscript.

Minor comments:

Line 27, "growth"

Corrected (now line 27)

Line 38, "intensified"

Corrected (now line 39)

Line 292, "grow on"

Corrected (now line 323)

Line 341, "either XUT1 permease"

Corrected (now line 377)

Line 374, "endocytosed"

Corrected (now line 404)

Line 393, "the"

Corrected (now line 424)

Reviewer 3 Report

Comment 1 – Line 15 – Abstract- Definition of hxt is missing.

Comment 2 – Line 26 – Abstract – What about SaXUT1?

Comment 3 – Line 27 – Abstract – sentence not ok.

Comment 4 – Line 30-32 – Abstract- This conclusion would be more supported if lysines were removed from both XUT1 enzymes.

Comment 5 – Lines 59-65 – A figure demonstrating these steps (the one from this study strain) would make the manuscript clearer.

Comment 6 – Line 73 – Sentence not ok.

Comment 7 – Line 88 – Define CRISPR-SpCas9

Comment 8 – Line 95 - How can HXT2 and HXT5 can be characterized in this strain if they are not deleted?

Comment 9 – Line 103 – In instead of with.

Comment 10 – line 120 - What are the auxotrophic requirements and concentrations used?

Comment 11 – line 132 - A660 nm definition should appear in line 128.

Comment 12 – line 135/146/212… – In should be L instead of l (liter).

Comment 13 – Line 146 – 25 mL of medium in 50 mL bottles? In line 130 it is mentioned that erlenmeyers were only filled to 1/5, why the big difference? Why the differences in the rotation?

Comment 14 – It should be S. arborariae, S. stipitis and S. passalidarium and not Sp. arboriae, Sc. Stipites and Sp. passalidarium, respectively.

Comment 15 – line 169 – Define these genes

Comment 16 – line 203 – Define DCW.

Comment 17 – general comment, proteins are generally not in italic. Only the part related to the microorganism’s name.

Comment 18 – Figure 1 - Biomass should be in g/L. How is iit possible to have an A600 of 15 in figure 1?

Comment 19 – line 276 - How was the ubiquitinylation potential determined in UbPred program? More information regarding this tool should be added. The website should be added. Add information in M and M section.

Comment 20 – lines 273 – 276- How can the differences be explained if the residues are in the same positions?

Comment 21 – lines 287-290 – It would be interesting to show the results of SaXUT1 with only C terminal truncation for comparison.

Comment 22 – line 293 – Grow instead of growth.

Comment 23 – line 294 – It is not clear why the authors consider that the consumption of xylose was efficient. The strain took almost 100 h to start to consume xylose and the consumption was very low. This does not seem very efficient. In addition, the growth was also very low in Figure 1. It is not clear why the discussion starts with figure 1 and that moves automatically for figure 4 without explaining the differences of both figures (different culture media??)

Comment 24 – Lines297-299 – The results obtained with modified SaXUT1 should be added to the manuscript (Figure). How the authors can be sure that the truncation only in C terminal would not improve the results? The results obtained with only C terminal truncation should be reported. N-terminal truncation may have originated additional problems.

Comment 25 – How is modified SpXUT1 compared to SsXUT1? Which one performs better?

Comment 26 – Lines 327 – It is not clear with the authors performed these tests only with wild-type proteins instead of the truncated versions.

Comment 27 – Figure 5 – Why the authors only presented growth results? Where are the results regarding substrate consumption and ethanol production?

Comment 28 – lines 350 – This is not consistent with the results; the double mutation did not improve growth.

Comment 29 – 353-354 – Results should be added (if not very relevant should be added to supplementary section).

Comment 30 – Abstract – The last sentence should be modified in order to not mislead the reader since that the approach where lysines are removed to improve functionality is not new, it is not a novelty, as the authors show in the discussion section.

Comment 31 - English should be carefully revised.

Author Response

Response to reviewer # 3.

We thanks the comments and suggestions raised by the reviewer, which we think contributed to improve the manuscript.

Comments and Suggestions for Authors

Comment 1 – Line 15 – Abstract- Definition of hxt is missing.

The text was modified accordingly (now line 15-16): “a Saccharomyces cerevisiae strain lacking the main sugar/hexose transporters (hxt-null)”

Comment 2 – Line 26 – Abstract – What about SaXUT1?

The statement is also valid to SaXUT1, although the performance of the strain expressing this permease was not as impressive as with the results obtained with the truncated SpXUT1 transporter (see answer to next comment).

Comment 3 – Line 27 – Abstract – sentence not ok.

The sentence was modified to: “While SsXUT1 lacks lysine residues in its N-terminal domain with high ubiquitinylation potential and has only one at the C-terminal domain, both SpXUT1 and SaXUT1 transporters have several of such residues at their C-terminal domains.” (now lines 23-26)

Comment 4 – Line 30-32 – Abstract- This conclusion would be more supported if lysines were removed from both XUT1 enzymes.

The results were also observed for SaXUT1 as depicted in now Fig. 6, new Table 3 and in the text (lines 331-334): “In the case of the SaXUT1 truncated in both the N- and C-terminal domains (SaXUT1ΔNΔC) we also observed a clear improvement in growth on glucose or xylose, but not as impressive as with the results obtained with the SpXUT1ΔC shown above.” The fermentation performance of the strain expressing the SaXUT1ΔNΔC transporter with glucose are shown in lines 334-338 and Supplementary Figure S1.

Comment 5 – Lines 59-65 – A figure demonstrating these steps (the one from this study strain) would make the manuscript clearer.

We thank the reviewer and a new Figure 1 with the xylose metabolizing pathways has been included in our revised manuscript (new Figure 1 in page 2)

Comment 6 – Line 73 – Sentence not ok.

The sentence was modified to: “Thus, contributes to improve of 2G ethanol production have been given via the heterologous expression of pentose transporters in S. cerevisiae.” (now line 75-76)

Comment 7 – Line 88 – Define CRISPR-SpCas9

The text was modified to: “using modern genome editing (CRISPR) technology” (now line 90-91).

Comment 8 – Line 95 - How can HXT2 and HXT5 can be characterized in this strain if they are not deleted?

Note that in the previous sentence of the manuscript (now lines 94-96) it was mentioned that the main “hexose transporters (hxt1Δ to hxt7Δ and gal2Δ)” were deleted, i.e. all between hxt1 and hxt7, including hxt2, hxt3, hxt4, hxt5, hxt6. The text in this part was also modified to improve clarity on the genetic background of the platform strain, as well as in the Materials and Methods section (lines 120-134).

Comment 9 – Line 103 – In instead of with.

Corrected as suggested (now line 108).

Comment 10 – line 120 - What are the auxotrophic requirements and concentrations used?

The description of the yeast media was improved: “Yeasts were grown in rich YP medium (1% yeast extract, 2% Bacto peptone, Sigma-Aldrich), synthetic complete (YNB) medium (0.67% yeast nitrogen base without amino acids, supplemented with 1.92 g/L of yeast synthetic Drop-out media without uracil, Sigma-Aldrich), or optimized [48] YNB medium (o-YNB, containing 1.34% yeast nitrogen base without amino acids, supplemented with 3.84 g/L of yeast synthetic Drop-out media without uracil and 1.5% casaminoacids, Sigma-Aldrich), with 2% maltose, glucose or xylose as carbon source.” (now lines 137-143)

Comment 11 – line 132 - A660 nm definition should appear in line 128.

The definition was added to the text: “optical density at 600 nm (A600nm)”, now line 148.

Comment 12 – line 135/146/212… – In should be L instead of l (liter).

Corrected (now lines 155, 166, 167, 233)

Comment 13 – Line 146 – 25 mL of medium in 50 mL bottles? In line 130 it is mentioned that erlenmeyers were only filled to 1/5, why the big difference? Why the differences in the rotation?

This setup is for fermentation experiments, where oxygen availability is not relevant for S. cerevisiae. The other experiments were designed to test growth under fully aerobic conditions. That is the reason for using 50 mL bottles (with low agitation) for fermentation, while growth was performed in erlenmeyers with higher agitation.

Comment 14 – It should be S. arborariae, S. stipitis and S. passalidarium and not Sp. arboriae, Sc. Stipites and Sp. passalidarium, respectively.

This is a standard nomenclature when we want to distinguish two different genus starting by the same letter. In order to distinguish the Spathaspora and Scheffersomyces we use Sp. and Sc. respectively (from the more common “S.” for Saccharomyces)

Comment 15 – line 169 – Define these genes

The following text was added: “two a–arrestins genes, the ROD1 and/or ROG3, known to be involved in sugar transporter endocytosis” (now lines 189-190)

Comment 16 – line 203 – Define DCW.

It is defined earlier in the text: “dry cell weight (DCW)/L)” (now line 165)

Comment 17 – general comment, proteins are generally not in italic. Only the part related to the microorganism’s name.

Corrected throughout the manuscript.

Comment 18 – Figure 1 - Biomass should be in g/L. How is iit possible to have an A600 of 15 in figure 1?

Biomass was measured either by dry cell weight (DCW) or by optical density (A600nm), both generally accepted.

Comment 19 – line 276 - How was the ubiquitinylation potential determined in UbPred program? More information regarding this tool should be added. The website should be added. Add information in M and M section.

This information was added in a separate section (2.5) in the Materials and Methods (lines 246-252).

Comment 20 – lines 273 – 276- How can the differences be explained if the residues are in the same positions?

The differences in ubiquitinylation potential of the lysine residues can be result of small sequence differences around each lysine in the cytoplasmic domain, and the output of the UbPred program is shown (the only difference is if they are with high or medium ubiquitinylation potential). Other programs can assign other degrees of ubiquitinylation potentials to these same lysine residues.

Comment 21 – lines 287-290 – It would be interesting to show the results of SaXUT1 with only C terminal truncation for comparison.

We agree with the reviewer, but unfortunately we do not have the SaXUT1 transporter truncated only in the C-terminal domain

Comment 22 – line 293 – Grow instead of growth.

Corrected (now line 323)

Comment 23 – line 294 – It is not clear why the authors consider that the consumption of xylose was efficient. The strain took almost 100 h to start to consume xylose and the consumption was very low. This does not seem very efficient. In addition, the growth was also very low in Figure 1. It is not clear why the discussion starts with figure 1 and that moves automatically for figure 4 without explaining the differences of both figures (different culture media??)

The authors agree that it is not efficient and rephrased it accordingly: “Nevertheless, the strain expressing the SpXUT1ΔC permease was able to ferment xylose, producing both ethanol and xylitol, while the strain expressing the full-length transporter (SpXUT1) was unable to metabolize the pentose as described above (Figure 5).” (lines 325-328). While now Figure 2 shows growth under fully aerobic conditions (poor growth on xylose), now Figure 5 (old Figure 4) shows fermentation of xylose under high cell density conditions (and low aeration/agitation), which occurred efficiently (even better that the strain expressing the previously characterized SsXUT1 permease [27], see comment in lines 328-331).

Comment 24 – Lines297-299 – The results obtained with modified SaXUT1 should be added to the manuscript (Figure). How the authors can be sure that the truncation only in C terminal would not improve the results? The results obtained with only C terminal truncation should be reported. N-terminal truncation may have originated additional problems.

We have added Supplementary Figure S1 with the relevant results of xylose fermentation by the SaXUT1 and truncated SaXUT1 transporters. We agree with the reviewer’s comments, but unfortunately we do not have the SaXUT1 transporter truncated only in the C-terminal domain.

Comment 25 – How is modified SpXUT1 compared to SsXUT1? Which one performs better?

The truncated SpXUT1 performed better than the previously characterized SsXUT1 (see lines 328-331)

Comment 26 – Lines 327 – It is not clear with the authors performed these tests only with wild-type proteins instead of the truncated versions.

To test the influence of the deleted arrestins we used the wild-type proteins (which did not work), and not the truncated versions that had already an improved performance (and thus probably hard to see if the absence of the arresting had some effect on the activity of the permease).

Comment 27 – Figure 5 – Why the authors only presented growth results? Where are the results regarding substrate consumption and ethanol production?

Besides the growth curves shown in now Figure 6, we are now presenting the ethanol produced at the end of the experiments (new Table 3), as well as the results of shake flasks experiments (see Supplementary Figure S2) performed with the DLG-K1ΔR3 strain (rog3 deleted) expressing both XUT1 transporters on glucose medium (the most relevant data).

Comment 28 – lines 350 – This is not consistent with the results; the double mutation did not improve growth.

To improve clarity the text was rephrased: “However, ROD1 and/or ROG3 knockout did not improve the utilization of xylose in the cells expressing the SpXUT1 permease, suggesting that other a–arrestins might be involved in ubiquitinylation of this transporter in the presence of the pentose.” (see lines 380-383)

Comment 29 – 353-354 – Results should be added (if not very relevant should be added to supplementary section).

The results are presented in Supplementary Figure S2.

Comment 30 – Abstract – The last sentence should be modified in order to not mislead the reader since that the approach where lysines are removed to improve functionality is not new, it is not a novelty, as the authors show in the discussion section.

The last sentence was modified accordingly to: “Taken together, these results suggest that the efficient heterologous expression of sugar transporters in S. cerevisiae requires the removal of lysines involved in ubiquitinylation and endocytosis.”. Note that the literature does not report a systematic study in relation to effect of lysines in the N- and/or C- terminus in the heterologous expression of sugar transporters in S. cerevisiae. The impact of the deletion of one of the a–arrestin genes further support relevance of the mechanism described and how can we use this approach for the efficient expression of heterologous transporters in S. cerevisiae.

Comment 31 - English should be carefully revised.

The text was extensively revised and detected grammatical errors corrected.

Round 2

Reviewer 1 Report

The manuscript has been improved after revision. But two issues remain. 

  1. Although N-terminal truncation of these heterologous transporters led to functional expression while then full-length proteins seemed to be not functional in S. cerevisiae, this strategy is not new. As mentioned in the manuscript, reference 48 reported N-terminal truncation to remove lysines from two heterologous cellubiose transporters.  The authors are suggested to revise the manuscript title to avoid this issue.
  2. The role of the N-terminal lysines in the transporter activity seems to be the main highlight of the work. More experimental evidence or at least more discussion would be helpful.

Author Response

We thanks again the comments and suggestions raised by the reviewer, which certainly contributed to improve the manuscript. We have also made more improvements in the English text.

The manuscript has been improved after revision. But two issues remain. 

  1. Although N-terminal truncation of these heterologous transporters led to functional expression while then full-length proteins seemed to be not functional in S. cerevisiae, this strategy is not new. As mentioned in the manuscript, reference 48 reported N-terminal truncation to remove lysines from two heterologous cellubiose transporters. The authors are suggested to revise the manuscript title to avoid this issue.

First, we would like to clarify that our results indicate that the truncation of the C-terminal lysine residues allow the functional expression of the XUT1 permeases, not the N-terminal truncation. The N-terminal truncation worked with the endogenous HXT1 permease from S. cerevisiae (ref. 68 and 73). The C-terminal truncation improved the stability of the heterologous cellobiose transporters, as shown in reference 48 (Internalization of heterologous sugar transporters by endogenous α-arrestins in the yeast Saccharomyces cerevisiae), which used a broad title (“sugar transporters”). Thus, we have changed as suggested the title of the manuscript to “Strategies for efficient expression of heterologous monosaccharide transporters in Saccharomyces cerevisiae”, removing “New” and been more specific (monosaccharide instead of “sugar”), and hope that we are now avoiding such conflicts, as pointed out by the reviewer. In several other instances of the text “sugar” was replaced by “monosaccharide” (e.g. lines 16, 31-32, 343-344, 415….) to be more specific.

  1. The role of the N-terminal lysines in the transporter activity seems to be the main highlight of the work. More experimental evidence or at least more discussion would be helpful.

Again, our results highlight the role of the C-terminal lysine residues in the functional expression of the cloned permeases. Nevertheless, we agree with the reviewer and have improved our discussion (lines 453-461, and 477-480) suggesting other possible experimental approaches to better characterize the involvement of lysine residues in the cytoplasmic terminal sequences (as well as arrestins) in the functional expression of heterologous monosaccharide transporters in S. cerevisiae.

Reviewer 3 Report

Comment 1 – Several modifications to the manuscript were not highlighted as expected. Several provided lines were not ok.

Comment 2 – line 95 – SSXYL1, SSXYL2 and XKS1 must be defined.

Comment 3 – Figure 1 needs to be cited in the text.

Comment 4 – line 90 - CRISPR should be defined.

Comment 5 – The different requirements of oxygen in the different experiments should be explained in the text.

Comment 6 – line 175 – It is not clear why the word sequence was removed.

Comment 7 – Again, it would be important to show the results with only C-terminal truncation to determine with mutations led to the improvement. At least the need to performed this experiments should be discussed in the manuscript.

Comment 8 – line 377 – Is the sentence corrected properly? It should be mention that the double knock out did not improve the results, right?

Comment 9 – line 478 – what is o-YNB?

Author Response

Response to reviewer # 3.

We thanks again the comments and suggestions raised by the reviewer, which certainly contributed to improve the manuscript. We have also made more improvements in the English text.

Comment 1 – Several modifications to the manuscript were not highlighted as expected. Several provided lines were not ok.

We apologize for this problem, but we would like to say that in the “revised with tracking changes” file all modifications were shown marked in red. We saw indeed that the “latest version of the manuscript for revision” file is different, with some changes highlighted in yellow, other changes are still in red (or even blue), and many others were not evident/highlighted at all. We do not know the reason for this problem. We also had problems with the comments of the reviewers, as many line numbers pointed by them did not match any file that we had. Our responses (with line numbers) were based in the PDF file (without tracking changes) we submitted in our reply, but we are not sure if the reviewers had access to it! (the problem with the submission system is that the template file only puts line numbers if it is saved as PDF). The editorial office of Journal of Fungi needs to fix such issues….. Nevertheless, our responses to the reviewers are based in the PDF file with line numbers (and without tracking changes) that we are submitting, besides the “revised new with tracking changes” file. Please refer to this file (PDF revised new without tracking changes).

Comment 2 – line 95 – SSXYL1, SSXYL2 and XKS1 must be defined.

We have modify this part, and better defined each gene (now lines 98-100 in the PDF file “revised new without tracking changes” that we are submitting)

Comment 3 – Figure 1 needs to be cited in the text.

We apologize for this carelessness in the revised manuscript. Now Figure 1 is cited in lines 57, 60 and 100.

Comment 4 – line 90 - CRISPR should be defined.

The definition is now in line 93.

Comment 5 – The different requirements of oxygen in the different experiments should be explained in the text.

We agree and the different oxygen conditions are not only specified in the Material and Methods section (lines 159-161 and 169), but also in the titles of Fig. 2 (line 283), Fig. 5 (line 338) and Fig. 6 (line 371), line 377, as well as the titles of Supplementary Fig. S1 (line 493) and S2 (line 495) – see also the Supplementary Material.

Comment 6 – line 175 – It is not clear why the word sequence was removed.

We agree and the “sequence” word was reestablished (now line 180)

Comment 7 – Again, it would be important to show the results with only C-terminal truncation to determine with mutations led to the improvement. At least the need to performed this experiments should be discussed in the manuscript.

We agree with the reviewer and have improved our discussion (lines 453-461, and 477-480) suggesting other possible experimental approaches to better characterize the involvement of lysine residues in the cytoplasmic terminal sequences (as well as arrestins) in the functional expression of heterologous monosaccharide transporters in S. cerevisiae.

Comment 8 – line 377 – Is the sentence corrected properly? It should be mention that the double knock out did not improve the results, right?

We agree and have changed the sentence (now lines 366-369).

Comment 9 – line 478 – what is o-YNB?

“o-YNB”, which stands for “optimized YNB”, was defined previously in the Materials and Methods section (lines 140-141). To better characterize this medium, we change it to “O-YNB” (lines 141, 372, and 496). We hope that now is clearer.

Round 3

Reviewer 3 Report

The document was significantly improved.

Lines 457-458 (from manuscript with track changes) – The English should be improved.